# Oxygen-Plasma-Induced Hetero-Interface NiFe_2_O_4_/NiMoO_4_ Catalyst for Enhanced Electrochemical Oxygen Evolution

**DOI:** 10.3390/ma15103688

**Published:** 2022-05-20

**Authors:** Nuo Xu, Wei Peng, Lei Lv, Peng Xu, Chenxu Wang, Jiantao Li, Wen Luo, Liang Zhou

**Affiliations:** 1Department of Physics, School of Science, Wuhan University of Technology, Wuhan 430070, China; mr.promise@whut.edu.cn (N.X.); 245856@whut.edu.cn (C.W.); 2State Key Laboratory of Advanced Technology for Materials Synthesis and Processing, Wuhan University of Technology, Wuhan 430070, China; pw0225@whut.edu.cn (W.P.); lvlei1028@whut.edu.cn (L.L.); xupeng620@whut.edu.cn (P.X.); liangzhou@whut.edu.cn (L.Z.); 3Chemical Sciences and Engineering Division, Argonne National Laboratory, Lemont, IL 60439, USA; jiantao_work@126.com

**Keywords:** hetero-interface, oxygen plasma, oxygen evolution reaction, electrocatalysis, electronic modulation

## Abstract

The electrolysis of water to produce hydrogen is an effective method for solving the rapid consumption of fossil fuel resources and the problem of global warming. The key to its success is to design an oxygen evolution reaction (OER) electrocatalyst with efficient conversion and reliable stability. Interface engineering is one of the most effective approaches for adjusting local electronic configurations. Adding other metal elements is also an effective way to enrich active sites and improve catalytic activity. Herein, high-valence iron in a heterogeneous interface of NiFe_2_O_4_/NiMoO_4_ composite was obtained through oxygen plasma to achieve excellent electrocatalytic activity and stability. In particular, 270 mV of overpotential is required to reach a current density of 50 mA cm^−2^, and the overpotential required to reach 500 mA cm^−2^ is only 309 mV. The electron transfer effect for high-valence iron was determined by X-ray photoelectron spectroscopy (XPS). The fast and irreversible reconstruction and the true active species in the catalytic process were identified by in situ Raman, ex situ XPS, and ex situ transmission electron microscopy (TEM) measurements. This work provides a feasible design guideline to modify electronic structures, promote a metal to an active oxidation state, and thus develop an electrocatalyst with enhanced OER performance.

## 1. Introduction

Nowadays, the dramatically increased demand for fossil energy has resulted in the depletion of traditional energy materials and has generated concerns regarding energy security and the environmental pollution caused by the extensive use of fossil energy [1]. Overall, water electrolysis has developed as an ideal and effective approach for producing hydrogen, an alternative clean energy source to traditional fossil fuels [2,3]. The oxygen evolution reaction (OER) involves multiple steps of proton coupling and a complex four-electron transfer process [4,5,6]. The sluggish reaction kinetics eventually cause a high enough overpotential to trigger the OER, which is a key factor limiting the efficiency of water electrolysis [3,7]. To date, the most effective catalysts for the OER have been found to be rare metal oxides, such as IrO_2_ and RuO_2_, as they significantly reduce the overpotential required for the OER. However, the high expenditure and scarcity of precious metals restrict their wide application for practical industrialization as efficient electrocatalysts.

Due to the abundant transition metal resources of the Earth itself, the construction of non-noble metal OER electrocatalysts has received much attention recently [6,8,9,10]. This includes transition metal compounds based on nickel, iron, or cobalt (Ni, Fe, and Co nonoxides [11,12,13,14]; oxides [15,16,17,18]; hydroxides [19,20,21]; and oxyhydroxides [22,23,24,25]), which have shown high conversion efficiencies towards the OER as substitutes for a precious metal electrocatalyst. Meanwhile, growing nanostructured catalysts directly on conductive substrates, such as Ni foam, has been established to decrease contact resistance and effectively improve energy efficiency [26,27,28]. Among these transition metal compounds, NiMoO_4_ on Ni foam has been researched as a promising electrocatalyst for its facile synthesis in large quantities. However, its intrinsic activity still remains defective, especially for its deficient active site (only Ni site) [29,30,31,32].

Generally, it is crucial to consider the modification of structure and electronic configuration in order to achieve outstanding OER performance, especially in order to attain a higher current at a lower overpotential with long-term stability [5]. An atomic-scale approach for constructing a reliable interface, especially a hetero-interface between different nanomaterials, has been intensively adopted to modify the local electronic structure of materials [33]. This approach can accelerate the reaction kinetics by combining the structural advantages of each component, thus improving the catalytic performance of nanocomposites [34,35,36,37,38]. For example, Lv et al. synthesized a core–shell structure of NiFe-60/Co_3_O_4_ on Ni Foam with an obvious and clear hetero-interface between the Co_3_O_4_ nanowire and the NiFe-layered double hydroxide nanosheet [39]. A hetero-interface contributes to the interaction between two different nanomaterials, facilitates electron transfer, and further leads to enhanced catalytic activity for the OER. Zhang et al. demonstrated a CoN_4_-based metal−organic framework (MOF) with embedded CoFeO_x_ nanoparticles; Co sites anchoring on the CoFeO_x_/MOF interface brought about an altered 3D electronic configuration for the interfacial Co and a higher valence [40]. In addition, composites consisting of multivariate transition metals can promote the exceptional modification of active sites within the matrix, and thus improve reaction efficiency and durability [41]. Based on this, the addition of Fe elements has been confirmed as a proper approach to enrich the active sites and boost highly efficient OER performance [42,43,44].

Herein, we report a hetero-interface made of NiFe_2_O_4_ nanoparticles (NPs) and a NiMoO_4_ nanowire (denoted as NiFe_2_O_4_/NiMoO_4_). Briefly, NiMoO_4_ nanowires were prepared on nickel foam through a facile hydrothermal synthesis. NiFe Prussian blue analogs (NiFe PBA) were firstly fixed on the NiMoO_4_ nanowires by iron exchange. Then, oxygen (O_2_) plasma converted the NiFe PBA to NiFe_2_O_4_ to form a NiFe_2_O_4_/NiMoO_4_ hetero-interface. As-synthesized, the NiFe_2_O_4_/NiMoO_4_ exhibits excellent performance for the OER with a low overpotential of 309 mV required to reach 500 mA cm^−2^ and a satisfactory stability (a 4% increase in the overpotential at 50 mA cm^−2^ over 150 h). The shift in the binding energy of metal sites increased the electronic interaction of the modulated hetero-interface. To understand its excellent OER performance, in situ Raman measurements, ex situ transmission electron microscopy (TEM), and ex situ X-ray photoelectron spectroscopy (XPS) were used to confirm the fast and irreversible reconstruction and identify the true active species in the catalytic process.

## 2. Results and Discussion

The schematics shown in Figure 1a illustrate the preparation of NiFe_2_O_4_ nanoparticles integrated with NiMoO_4_ nanowires on nickel foam. Briefly, through a simple hydrothermal method [45], hydrated NiMoO_4_ nanowires were vertically germinated on Ni foam. In accordance with a previous report [46], the NiFe PBA was grown on NiMoO_4_ nanowires and the MoO_4_^2−^ on the surface of the nanowires was replaced with K^+^ and [Fe(CN)_6_]^3+^ by ion exchange. The NiFe PBA on the NiMoO_4_ surface was converted to NiFe_2_O_4_ NPs under O_2_ plasma treatment, and NiFe_2_O_4_/NiMoO_4_ was obtained. For comparison, NiMoO_4_ nanowires were also placed under the same O_2_ plasma treatment and denoted as NiMoO_4_ O_2_-Pl.

**Characterization of NiMoO_4_**. Appendix A indicate that the NiMoO_4_ possessed an even and well-defined nanowire morphology. Its average diameter was about 100 nm. As the XRD patterns show in Appendix A, the diffraction peaks of the NiMoO_4_ were in perfect agreement with the NiMoO_4_·xH_2_O (JCPDF: 00-013-0128), which means the NiMoO_4_·xH_2_O was highly crystalline. In the FT-IR spectra shown in Appendix A, the two peaks at 1628 and 3446 cm^−1^ correspond to the stretching vibration of hydroxyl (-OH), which can be ascribed to the bending mode of crystal water in the NiMoO_4_·xH_2_O and the surface-adsorbed water molecules [47].

**Characterization of NiFe PBA/NiMoO_4_**. As shown in Figure 1b, a weak diffraction peak appears at 17.3°, which can be attributed to the KNi[Fe(CN)_6_] (JCPDF: 01-089-8978). At the same time, the diffraction peaks of the NiMoO_4_·xH_2_O still remain in the NiFe PBA/NiMoO_4_. In Appendix A, the surface of the NiMoO_4_ nanowires is covered with small-sized NiFe PBA NPs, indicating the expected process of the iron exchange. Figure 1c reveals peaks at 357, 828, 872, and 950 cm^−1^ for the NiFe PBA/NiMoO_4_, which can be assigned to the Mo-O vibration, and this result is consistent with previous reports [32]. In addition, the NiFe PBA/NiMoO_4_ exhibits three peaks around 2100 cm^−1^, which can be attributed to -CN [48,49]. In Appendix A, for the NiFe PBA/NiMoO_4_, a new characteristic peak can be observed at 2099 cm^−1^ in the FT-IR spectrum, which is attributed to the stretching vibrations of -CN in the NiFe PBA NPs [50]. In Appendix A, for the NiFe PBA/NiMoO_4_, the Ni 2p spectra can be deconvoluted into four peaks and two wide satellite peaks. In the Ni 2p_3/2_, the peaks at 856.2 eV and 857 eV can be ascribed to the Ni^2+^ and Ni^3+^ species, respectively. Meanwhile, in the Ni 2p_1/2_, the peaks of the Ni^2+^ and Ni^3+^ species can be fitted at 874.0 eV and 875.2 eV, respectively. In addition, two satellite peaks for Ni can be observed at 862.8 and 880.7 eV [51]. As shown in Appendix A, the further fitted peaks at 708.5 and 709.1 eV in the Fe 2p_3/2_ are ascribed to Fe^2+^ and Fe^3+^, respectively, while the peaks observed at 721.6 and 722.2 eV in the Fe 2p_1/2_ are owed to the existence of Fe^2+^ and Fe^3+^, respectively. The ratio of the Fe^3+^ to Fe^2+^ peak areas in the Fe 2p_3/2_ was calculated as 0.948. Furthermore, one satellite peak for Fe appears at 715.7 eV [51].

**Characterization of NiFe_2_O_4_/NiMoO_4_**. In Figure 1b, compared to the NiFe PBA/NiMoO_4_, the diffraction peak initially attributed to KNi[Fe(CN)_6_] disappears, and a new weak diffraction peak can be observed at 43.3°, which is attributed to the (400) planes of the NiFe_2_O_4_ (JCPDS: 44-1485). As shown in Figure 1c, the characteristic Raman peaks attributed to the Mo-O bond still remain, while the peak owed to -CN disappears. A broad peak at 520 cm^−1^ in Appendix A indicates the formation of NiFe-O. The same phenomenon is shown in Appendix A, as the characteristic peak of -C≡N- disappears and an apparent peak at 1384 cm^−1^ can be assigned to the C=O in CO_2_ and the stretching vibration of the interlayer NO_3_^=^ groups [47]. This is related to the decomposition of PBA under O_2_ plasma treatment. As shown in Figure 2a, the NiMoO_4_ remains in the structure of the nanowire with a diameter of 100 nm, similarly to the NiMoO_4_ and NiFe PBA/NiMoO_4_, while the NiFe_2_O_4_ nanoparticles slightly agglomerate. The TEM image of the NiFe_2_O_4_/NiMoO_4_ (Figure 2b) clearly shows the NiFe_2_O_4_ NPs anchoring on the surface of the NiMoO_4_ nanowire. The high-resolution transmission electron microscope (HRTEM) image in Figure 2c indicates the high-crystalline characteristic of the NiMoO_4_. The HRTEM image in Figure 2d shows the apparent hetero-interface of the NiFe_2_O_4_/NiMoO_4_. The lattice spacing of 2.08 Å can be attributed to the (400) plane of the NiFe_2_O_4_ and the lattice spacing of 3.26 Å assigned to the NiMoO_4_·xH_2_O. The elemental mapping images (Figure 2e) indicate that the Fe element is evenly distributed on the NiMoO_4_ nanowires in the form of nanoparticles. Appendix A shows the chemical composition of the NiFe_2_O_4_/NiMoO_4_. The molecular ratio of NiFe_2_O_4_ and NiMoO_4_ in the NiFe_2_O_4_/NiMoO_4_ is 1:17.27.

XPS was used to analyze and further explore the surface electronic interaction of the NiFe_2_O_4_/NiMoO_4_. The Ni 2p spectra of NiFe_2_O_4_/NiMoO_4_ contain four fitted peaks with wide satellites (Figure 3a). The fitted peaks at 856.1 and 873.7 eV in the Ni 2p_3/2_ and the Ni 2p_1/2_, respectively, can be attributed to Ni^2+^, while another two peaks (858.3 and 875.9 eV) correspond to the Ni^3+^ species. The wide peaks observed at 862.3 and 879.9 eV are owed to the satellites of Ni [52]. As Figure 3b shows, the fitted peaks at 710.7 and 713.3 eV can be related to Fe^2+^ and Fe^3+^ in the Fe 2p_3/2_, and the same is true for another two peaks at 723.8 and 726.4 eV in the Fe 2p_1/2_ [53]. The broad peaks at 718.8 and 731.9 eV can be attributed to the satellite peaks of Fe. In Appendix A, the peaks in the Ni 2p_3/2_ and the Mo 3D of the NiFe_2_O_4_/NiMoO_4_ exhibit slightly negative shifts of about 0.2 eV compared with those observed from the spectra of the NiFe PBA/NiMoO_4_. The peaks of the Fe 2p exhibit a distinct positive shift compared with those of the NiFe PBA/NiMoO_4_, and the ratio of the Fe^3+^ to Fe^2+^ peak area in the Fe 2p_3/2_ (Appendix A and Figure 3b) increases from 0.948 in the NiFe PBA/NiMoO_4_ to 1.706 in the NiFe_2_O_4_/NiMoO_4_. The negative movement of binding Ni and Mo energy indicates the regulation of the electronic structure in the hetero-interface. Meanwhile, oxygen plasma leads to the oxidation of Fe, and these two factors eventually promote an increase in the binding energy of Fe. It has been confirmed that Fe with a high valence state promotes processes in the OER [54,55,56]. The fitted O 1s peaks at 530.7, 532.3, and 533.3 eV can be attributed to metal-O, C=O [57], and surface-adsorbed oxygen, respectively.

**Electrochemical performance.** For the purpose of measuring the electrochemical performance of the prepared samples, a three-electrode electrochemical system was used. An aqueous solution of 1 M KOH was selected as the electrolyte solution. The polarization curves of all samples with *iR* corrected are shown as Figure 4a. The peaks around 1.38 V can be assigned to the oxidation of nickel species to a higher valence state. Furthermore, the NiFe_2_O_4_/NiMoO_4_ demonstrates the lowest overpotential of 253 mV to reach 10 mA cm^−2^, while the overpotential required to achieve 10 mA cm^−2^ for the NiFe PBA/NiMoO_4_, NiMoO_4_ O_2_-Pl, NiMoO_4_, and Ni foam is 310, 313, 324, and 431 mV, respectively. In addition, for the NiFe_2_O_4_/NiMoO_4_, an overpotential of 270 and 309 mV are required to achieve 50 mA cm^−2^ and 500 mA cm^−2^, respectively.

As shown in Figure 4b, the Tafel slope of the NiFe_2_O_4_/NiMoO_4_ is the smallest, at 46.4 mV dec^−1^, compared with that of the NiFe PBA/NiMoO_4_ (119.2 mV dec^−1^), NiMoO_4_ O_2_-Pl (139.8 mV dec^−1^), NiMoO_4_ (136.8 mV dec^−1^), and Ni foam (230.1 mV dec^−1^). The smaller Tafel slope of the NiFe_2_O_4_/NiMoO_4_ indicates its faster kinetics [4,6]. The high performance of the NiFe_2_O_4_/NiMoO_4_ can be attributed to the oxygen-plasma-induced formation of the hetero-interface, made up of NiFe_2_O_4_ NPs and NiMoO_4_ nanowire arrays and containing iron with a higher valence. Iron with a higher valence has been confirmed to be conducive to the OER [54,55,56,57].

Figure 4c shows the electrochemical impedance spectroscopy (EIS) of all samples, and it can clearly be observed that the smallest charge transfer resistance (R_ct_) is found in the NiFe_2_O_4_/NiMoO_4_. The smaller R_ct_ relative to the others indicates a faster charge transfer for the NiFe_2_O_4_/NiMoO_4_, which may relate to the hetero-interface made up of NiFe_2_O_4_ NPs and NiMoO_4_ and further leads to enhanced electrocatalytic performance.

The electrochemical active surface area (ECSA) by CV measurement is shown in Appendix A. As shown in Figure 4d, the double-layer capacitance (C_dl_) of the NiFe_2_O_4_/NiMoO_4_, NiFe PBA/NiMoO_4_, NiMoO_4_ O_2_-Pl, and NiMoO_4_ was calculated to be 4.21, 3.09, 2.49, and 3.67 mF cm^−2^. The larger value of C_dl_ indicates a higher electrocatalytic OER activity of the NiFe_2_O_4_/NiMoO_4_, which is attributed to more exposed active sites related to the iron with a higher valence. Long-term stability is an important index for evaluating a catalyst. As shown in Figure 4e, the NiFe_2_O_4_/NiMoO_4_ displays outstanding durability (a 4% increase in the overpotential at 50 mA cm^−2^ over a 150 h reaction).

**In situ Raman spectra.** To figure out the phase change and reconstruction in the OER process, the NiFe_2_O_4_/NiMoO_4_ was first activated in an alkaline solution. In Figure 5a, with the increase in the applied potential from 1.18 V to 1.43 V, the intensity of characteristic peaks for Mo-O vibration decreased, which represents the dissolution of MoO_4_^2−^. Meanwhile, a small but sharp characteristic peak at 525 cm^−1^ corresponding with the Fe-O bond in FeOOH emerged with an applied potential of 1.23 V [58], which indicates the formation of FeOOH in the OER process. When the potential is applied at 1.28 V, broad peaks can be observed around 460 cm^−1^ and 520 cm^−1^, which are related to the appearance of α-Ni(OH)_2_ [59]. The peak becomes sharper when the applied potential arrives at 1.33 V. A broad peak occurs at 475 cm^−1^, which can be attributed to the emergence of γ-NiOOH from α-Ni(OH)_2_ [32,58], and the peak tends to become sharper with an applied potential at 1.43 V, while another characteristic peak of γ-NiOOH appears at 558 cm^−1^ [32,59].

In situ Raman spectra of the NiFe_2_O_4_/NiMoO_4_ in the initial two cycles in CVs are shown in Figure 5b. With multiple cycles, the intensity of the characteristic peak for γ-NiOOH gradually stabilizes and the characteristic peaks of Mo-O vibration almost completely disappear, which can be attributed to the irreversible reconstruction of the NiFe_2_O_4_/NiMoO_4_.

For comparison, the NiFe PBA/NiMoO_4_ was also first activated in an alkaline solution. In Appendix A, with the increase in applied potential, the same phenomenon of a decrease in the intensity of characteristic peaks for Mo-O vibration can be observed. In addition, when a potential of 1.38 V is applied to the NiFe_2_O_4_/NiMoO_4_, a broad peak occurs at 475 cm^−1^, which can be attributed to the emergence of γ-NiOOH. The same phenomenon occurs at an applied potential of 1.28 V for NiFe_2_O_4_/NiMoO_4_. This fact means that the NiFe_2_O_4_/NiMoO_4_ is reconstructed faster than the NiFe PBA/NiMoO_4_, which leads to the better OER performance of the NiFe_2_O_4_/NiMoO_4_ from another aspect. However, there are no observable peaks attributed to FeOOH, and in Appendix A, with the increase in applied potential, the peaks related to -CN still exist [48,49]. This illustrates that the Fe coordinating with the cyanide group cannot catalyze the OER as an independent active site with the increase in applied potential, which further explains the reason that the OER performance of the NiFe PBA/NiMoO_4_ is close to the OER performance of the NiMoO_4_.

**Ex situ XPS.** The Ni 2p and Fe 2p spectra of the NiFe_2_O_4_/NiMoO_4_ after 3 h of OER testing are shown in Figure 3d,e, respectively. In Figure 3d, the Ni 2p can be deconvoluted into two peaks with two satellites. The fitted peaks at 855.1 and 872.7 eV can be ascribed to Ni^3+^, which is attributed to NiOOH [60]. Meanwhile, two satellites of Ni can be observed at 860.9 and 878.9 eV, respectively. As shown in Figure 3e, the two fitted peaks occur at 712.1 and 725.2 eV with a broad satellite at 718.3 eV, which can be related to FeOOH [61,62]. It can clearly be observed that there is a sharp attenuation in the peak intensity of the Mo 3d of the NiFe_2_O_4_/NiMoO_4_ after 3 h of OER testing, further demonstrating the irreversible reconstruction of the NiFe_2_O_4_/NiMoO_4_ with the dissolution of MoO_4_^2−^.

**Ex situ TEM.** The images of the NiFe_2_O_4_/NiMoO_4_ after OER testing (Figure 6a,b) clearly show the robust surface of the nanowire and numerous defects as the result of the dissolution of MoO_4_^2−^. The HRTEM image in Figure 6c reveals clear lattice fringes of the (105) plane for NiOOH (JCPDF: 00-006-0075) with a crystalline interplanar spacing of 2.09 Å. The HRTEM image in Figure 3d shows small black particles distributed in clumps, which may relate to the amorphous FeOOH delivered by the activation of the NiFe_2_O_4_ in OER testing. The elemental mapping images (Figure 3e) indicate that Fe is still evenly distributed on the NiMoO_4_ nanowire, and Mo dissolves in large quantities, which is consistent with the aforementioned analytical results.

## 3. Conclusions

In summary, a heterogeneous interface of NiFe_2_O_4_/NiMoO_4_ with high-valence iron through oxygen plasma can be fabricated to achieve excellent electrocatalytic activity and stability. To achieve a current density of 50 mA cm^−2^, 270 mV of overpotential is required, while an overpotential of 309 mV is required to reach 500 mA cm^−2^. The NiFe_2_O_4_/NiMoO_4_ also exhibits a satisfactory stability (a 4% increase in the overpotential at 50 mA cm^−2^ over 150 h). O_2_-plasma-induced electronic interaction in the hetero-interface of NiFe_2_O_4_/NiMoO_4_ and iron with a higher valence play an essential role in OER performance. The potential-dependent phase change and the fast and irreversible reconstruction of the NiFe_2_O_4_/NiMoO_4_ in a catalytic process were identified by in situ Raman, ex situ XPS, and ex situ TEM measurements. Based on this, the true active species, NiOOH and FeOOH, were determined. This work provides a feasible design guideline for modifying electronic structure through the construction of a heterogeneous interface and the activation of metal sites by O_2_ plasma, finally leading to enhanced OER performance.

## Figures and Tables

**Figure 1 materials-15-03688-f001:**
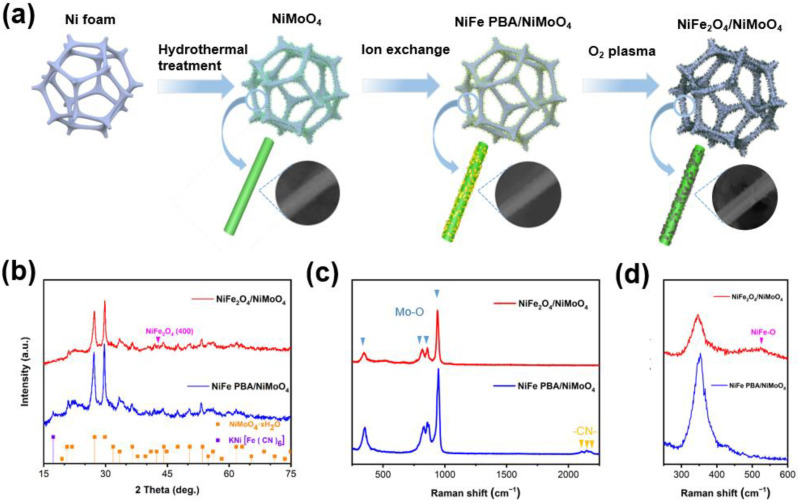
(**a**) Schematic illustration of the synthesis of NiFe_2_O_4_/NiMoO_4_; (**b**) XRD patterns of NiFe PBA/NiMoO_4_ and NiFe_2_O_4_/NiMoO_4_; (**c**) Raman spectra of NiFe PBA/NiMoO_4_ and NiFe_2_O_4_/NiMoO_4_; and (**d**) Raman spectra of NiFe PBA/NiMoO_4_ and NiFe_2_O_4_/NiMoO_4_ in a region from 250 cm^−1^ to 750 cm^−^^1^, respectively.

**Figure 2 materials-15-03688-f002:**
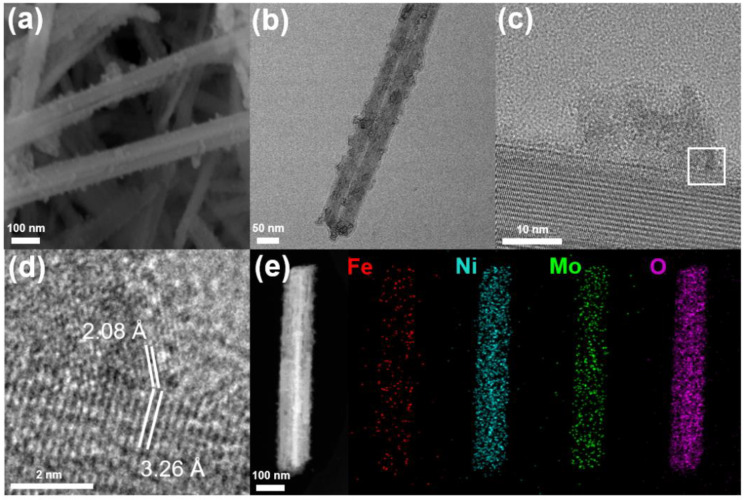
(**a**) SEM image, (**b**) TEM image, and (**c**) HRTEM images of NiFe_2_O_4_/NiMoO_4_; (**d**) the corresponding HRTEM images of selected areas; and (**e**) EDS mapping images for Fe, Ni, Mo, and O elements of NiFe_2_O_4_/NiMoO_4_.

**Figure 3 materials-15-03688-f003:**
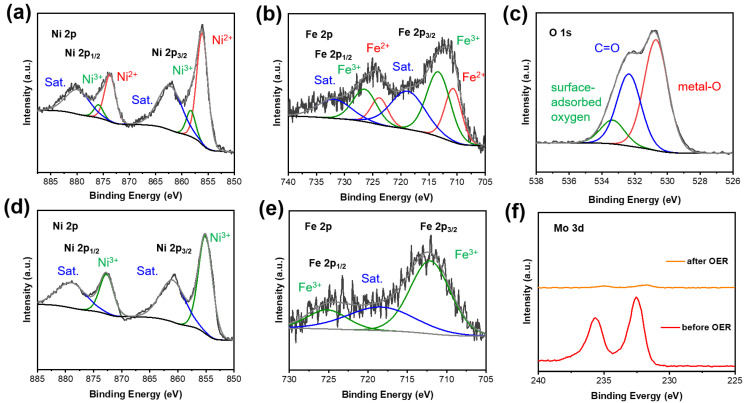
XPS: (**a**) Ni 2p, (**b**) Fe 2p, and (**c**) O 1s spectra of NiFe_2_O_4_/NiMoO_4_; (**d**) Ni 2p and (**e**) Fe 2p spectra of NiFe_2_O_4_/NiMoO_4_ after OER testing for 3 h; (**f**) Mo 3D spectra of NiFe_2_O_4_/NiMoO_4_ before OER testing and after OER testing.

**Figure 4 materials-15-03688-f004:**
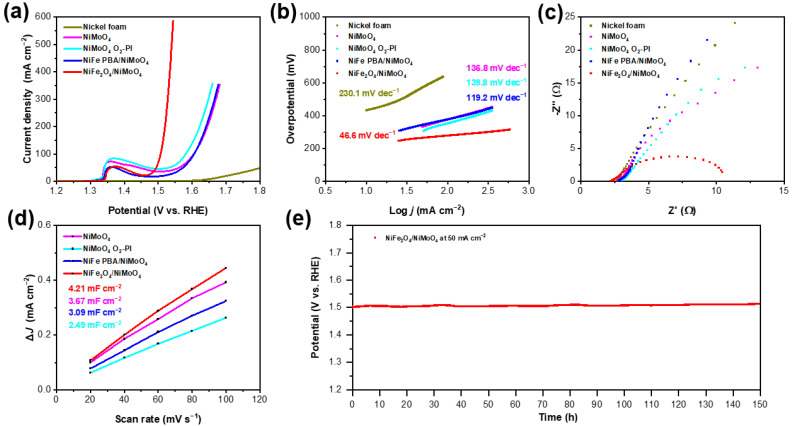
(**a**) Polarization curves and (**b**) corresponding Tafel slope plots of as-prepared catalysts; (**c**) EIS Nyquist plots of NiMoO_4_, NiMoO_4_ O_2_-Pl, NiFe PBA/NiMoO_4_, and NiFe_2_O_4_/NiMoO_4_; (**d**) capacitive current densities plotted as a function of the scan rate; and (**e**) chronopotentiometry of NiFe_2_O_4_/NiMoO_4_ at 50 mA cm^−2^ with *iR* corrected.

**Figure 5 materials-15-03688-f005:**
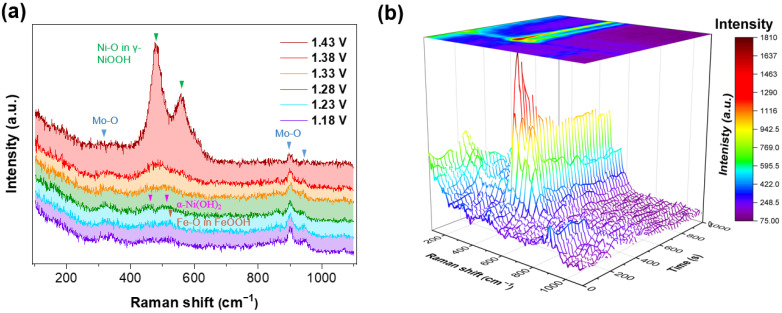
In situ Raman spectra of NiFe_2_O_4_/NiMoO_4_ (**a**) for activation from 1.18 V to 1.43 V and (**b**) for CVs in the initial 2 cycles from 1.18 V to 1.43 V at a scan rate 1 mV s^−1^.

**Figure 6 materials-15-03688-f006:**
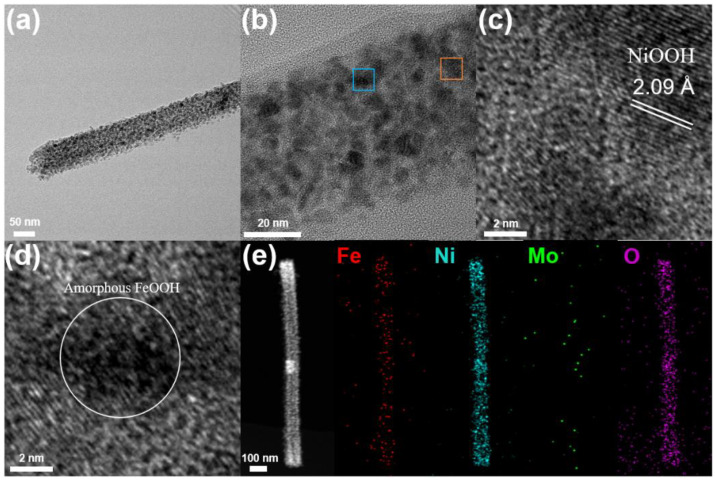
(**a**) TEM image; (**b**) HRTEM image of NiFe_2_O_4_/NiMoO_4_ after 3 h of OER testing; (**c**) the corresponding HRTEM images of the area selected by the orange frame; (**d**) the corresponding HRTEM images of the area selected by the blue frame; and (**e**) EDS mapping images for Fe, Ni, Mo, and O elements of NiFe_2_O_4_/NiMoO_4_ after OER.

## Data Availability

The data presented in this study are available from the corresponding author, upon reasonable request.

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
