# Peer review of "Oxygen-Plasma-Induced Hetero-Interface NiFe2O4/NiMoO4 Catalyst for Enhanced Electrochemical Oxygen Evolution"

_materials, 2022, doi:10.3390/ma15103688_

Round 1

Reviewer 1 Report

Reviewer comments:

The research manuscriptOxygen plasma induced hetero-interface NiFe2O4/NiMoO4 catalyst for enhanced electrochemical oxygen evolution” can be considered for publication after major revision

  • The language is not smooth and lot of grammatical and typographical errors needs to be corrected throughout the manuscript.
  • Nanoparticles has been spelled as nanoparticals throughout the manuscript!. Correct it

Valence instead valance

  • In the FT-IR spectra, peak at 3446 cm−1 can be assigned for stretching vibration of –OH and the peak at 1628 cm-1 is not assigned for –OH. Check it!
  • NiFe2O4/NiMoO4, has lower overpotential for OER with lower Tafel slope value ? But it is mentioned in the text that NiFe PBA/NiMoO4 has smallest Tafel slope value of 46.4 194 mV dec−1. Correct it
  • In general, hetero-interface increases the Rct value but it has decreased the Rct value here! Why?
  • Provide the mechanism for OER.
  • Improve the Introduction by adding organic macrocycles and other metal organic frameworks. For example,

Sustainable Energy and Fuels, 5, 1448, 2021

Journal of Power Sources, 449, 227516, 2020

Reviewer 2 Report

  1. The authors did a good work. However, what is the novelty of this?
  2.  

Reviewer 3 Report

This paper describes the synthesis of oxygen plasma induced hetero-interface NiFe2O4 /NiMoO4 catalyst that exhibited enhanced electrochemical oxygen evolution (270 mV overpotential is required to reach a current density of 50 mA cm-2). They described that the heterogeneous interface of NiFe2O4/NiMoO4 with high-valence iron through oxygen plasma is fabricated to achieve excellent electrocatalytic activity and stability, and O2 plasma induced electronic interaction in hetero-interface NiFe2O4/NiMoO4 and iron with higher valence play an essential role in OER performance. The presented results are interesting and is suitable for publication in this journal after revision according to the following comments.

  1. Low magnified TEM or SEM image of NiFe2O2/NiMoO4 should be presented to understand that the schematic image shown in Figure 1a is reasonable.
  2. Error on Page 5: The sentence “…the Tafel slope of the NiFe PBA/NiMoO4 is smallest of 46.4 mV dec−1….” is wrong. In the sentence, “NiFe PBA/NiMoO4” should be corrected to “NiFe2O4 /NiMoO4”.
  3. Errors in References: (i) Ref. 31: “Journal of Materials Chemistry A” should be corrected to “J. Mater. Chem. A”; (ii) Ref. 41: Delete **; (iii) Journal names are missing for Refs. 54, 57, 59, 60, 61. The authors should be careful in preparation of the manuscript.
  4. Describe the ratio of NiFe2O2 and NiMoO4 in NiFe2O2/NiMoO4.
  5. DFT study is preferred to clarify the interaction of Ni and Mo in NiFe2O2/NiMoO4 leading to the high OER.
  6. Describe ORR activity of NiFe2O4 /NiMoO4 and NiFe PBA/NiMoO4 for comparison. Some spinel catalysts have been reported to show bifunctional (ORR and OER) activities and compare to the reported results (for examples, see Yang et al. ChemCatChem, 2019, 11, 5929-5944;  Rep., 2017, 7, 45384).

Round 2

Reviewer 1 Report

The research manuscript may be accepted for publication